# Performance of Sprayed PVDF-Al_2_O_3_ Composite Coating for Industrial and Civil Applications

**DOI:** 10.3390/ma14216358

**Published:** 2021-10-24

**Authors:** Adel M. A. Mohamed, Hosam Hasan, Mohamed M. El-Sayed Seleman, Essam Ahmed, Sayed M. Saleh, Rehab M. El-Maghraby

**Affiliations:** 1Department of Metallurgical and Materials Engineering, Faculty of Petroleum and Mining Engineering, Suez University, Suez 43512, Egypt; hosam.hasan@suezuniv.edu.eg (H.H.); mohamed.elnagar@suezuniv.edu.eg (M.M.E.-S.S.); essam.ahmed@suezuniv.edu.eg (E.A.); 2Department of Chemistry, College of Science, Qassim University, Buraidah 51452, Saudi Arabia; e.saleh@qu.edu.sa; 3Chemistry Branch, Department of Science and Mathematics, Faculty of Petroleum and Mining Engineering, Suez University, Suez 43512, Egypt; 4Department of Petroleum Refinery and Petrochemical Engineering, Faculty of Petroleum and Mining Engineering, Suez University, Suez 43512, Egypt; rehab.elmaghraby@suezuniv.edu.eg

**Keywords:** superhydrophobicity, PVDF, Al_2_O_3_ nanoparticles, composite, morphology, corrosion, nanoindentation

## Abstract

Because of their great water repellency, Superhydrophobic coatings have a major impact on a variety of industrial applications. The current study’s key originality is the development of low-cost, stable, superhydrophobic, and corrosion-resistant composite coatings. In the present work, polyvinylidene fluoride (PVDF)/Al_2_O_3_ composite coatings were produced using the spray technique to investigate the wettability and corrosion behavior of the coated materials for industrial and civil applications. PVDF was mixed with various concentrations of Al_2_O_3_ nanoparticles, and the mixture was sprayed onto steel, aluminum, and glass substrates. The wettability and morphology of the coated surfaces were investigated using the sessile droplet method and scanning electron microscopy, respectively. The corrosion resistance of bare substrates was compared to that of those coated with PVDF alone and those coated with PVDF/Al_2_O_3_ nanoparticles using Tafel polarization techniques. The force of adhesion between the coat and the substrates was measured in pounds per square inch. A nanoindentation test was also used to measure the hardness of the coating layer. The PVDF/Al_2_O_3_ coated steel showed a significantly higher water contact angle and lower contact angle hysteresis, reaching 157 ± 2° and 7 ± 1°, respectively, compared to the coated aluminum and glass substrates. Corrosion test results showed that the superhydrophobic PVDF/Al_2_O_3_ coatings had a much higher corrosion protection efficiency for steel and aluminum than that of the PVDF ones. The PVDF/Al_2_O_3_ coated substrates showed moderate but still acceptable adhesion between the coating layer and the substrates. Moreover, the PVDF/Al_2_O_3_ coatings had much better mechanical properties than the PVDF only coatings. Such type of coating could be a promising candidate for possible industrial and civil applications.

## 1. Introduction

Because of their unique self-cleaning, anti-stick, and anticontamination, properties superhydrophobic surfaces (SHCs) have received a lot of attention [1]. Superhydrophobic coatings with a water contact angle (WCA) greater than 150° have received interest due to their water repellence, self-cleaning abilities, and corrosion resistance [2]. To date, many efforts have been devoted to the development of superhydrophobic coatings by forming a rough structure and/or reducing the surface energy by using low-surface energy materials. Because superhydrophobicity properties are a consequence of surface free energy and roughness, two methods for fabricating SHCs have been introduced: chemically altering a surface of a low-surface energy substance or enhancing the roughness of the target materials’ surface [3,4,5]. Chemical etching [6], electrochemical deposition [7], sol-gel [8], layer-by-layer assembly [9], plasma polymerization [10], and chemical vapor deposition [11] are some of the methods that can be used for the production rough surfaces with varied microstructures that have been reported thus far. Many techniques, such as chemical vapor transport and condensation [12]; pulsed laser deposition [13]; chemical vapor deposition [14]; and hydrothermal growth [15], have been used to produce materials with superhydrophobic surfaces that have a WCA of more than 150°. Yet, a great portion of these procedures entail stringent requirements (such as the use of hazardous chemicals), expensive ingredients, and complicated processing methods. Accordingly, a straightforward and simple technique that does not cost much or that has limits in the manufacture of superhydrophobic surfaces on a large scale should be widely employed.

Spray coating is a simple and financially accessible methodology for a wide scope of applications. It is not limited for application over a definite substrate and may be used over a vast surface area with ease; additionally, it rarely requires extra sophisticated or expensive application methods [16].

Polyvinylidene fluoride, which is originally a hydrophobic material, is regarded as an outstanding porous polymeric film and has been widely used in membrane distillation and membrane filtration due to its superior mechanical and thermal properties [17,18,19]. Although numerous approaches have been utilized to improve the hydrophobicity of PVDF to super levels [20,21], only a few superhydrophobic PVDF coatings have been recorded for use in industrial and civic applications. Chaoyi et al. [18] created a superhydrophobic PVDF coating on wind turbine blades using a unique and easy technique, resulting in WCA and sliding angle (SA) of 156 ± 1.9 and 2 degrees, respectively. Recently, inorganic nanoparticles including silica [22,23], titanium dioxide [12,24], alumina [25], zinc oxide [26,27], and zirconium dioxide [28,29] have recently been widely employed in polymers to improve the polymer properties to fit a specific commercial purpose. The incorporation of a higher proportion of inorganic components is a frequent aspect of these alterations. Surface roughness is linked to hydrophobicity and can be achieved by incorporating solid materials such as nickel, cerium, graphene, and metal oxides such as ZnO, SiO_2_, TiO_2_, and Al_2_O_3_ into the mix [30].

Al_2_O_3_ particles have gained a lot of attention among these inorganic materials because of their low surface energy, surface roughness, re-entrant structure, all of which are important for achieving hydrophobicity. In addition, its antibacterial properties; excellent mechanical, electrical insulation, and high-temperature properties; and first-rate impact, abrasion, and chemical resistance are desirable. Aluminum oxide (Al_2_O_3_) is one of the cheapest materials and has excellent usability in various applications. The use of Al_2_O_3_ in coatings has also attracted significant attention in recent years due to its antibacterial properties; excellent mechanical, electrical insulation, and high-temperature properties; and first-rate impact, abrasion, and chemical resistance properties. Although SiO_2_ is the most commonly used incorporated particle [31,32,33,34], Al_2_O_3_ has also been frequently utilized since it is tougher than SiO_2_ and is frequently used to improve scratch and abrasion resistance [35]. It is also more compatible with the organic solvents needed to generate the PVDF solution. Because of their small size, nanoscale particles differ from bulk materials in that they have a larger surface area. Several studies are available on the fabrication of a superhydrophobic Al_2_O_3_ coating on glass substrates for optically transparent, anti-reflective, and self-cleaning applications [36,37,38]; however, there are limited studies on PVDF–Al_2_O_3_ composite coatings. Therefore, the goal of this study was to investigate the wettability, morphology, corrosion, adhesion, and hardness of the prepared PVDF/Al_2_O_3_ composite coatings onto steel, aluminum, and glass substrates. The PVDF/Al_2_O_3_ coated steel/Al/glass substrates showed water-resistance and great mechanical stability.

## 2. Experimental Procedure

### 2.1. Starting Materials

Polyvinylidene fluoride (PVDF, (CH2 CF_2_)n) was provided by Sigma–Aldrich (Hamburg, Germany). PVDF solutions were supplemented with alumina (Al_2_O_3_) nanoparticles of <100 nm particle size and were purchased from Sigma–Aldrich (Hamburg, Germany). The solvents utilized were N, N-dimethylformamide (DMF, HCON(CH_3_)_2_, >99%, reagent), stearic acid, and hexane (C_6_H_14_), which were supplied by AlSAFWA Center (Cairo, Egypt).

### 2.2. Samples Preparation

Before coating, the steel, aluminum, and glass substrates were ultrasonically cleaned in acetone for 10 min and were then cleaned in distilled water for 10 min before being blown dry in a stream of air. To obtain a homogeneous solution, 5 g of PVDF was dissolved in DMF at 50 °C and was agitated at 600 rpm for 120 min. Different amounts of Al_2_O_3_ (1, 1.5, and 2 g) were dispersed in a hexane solution of stearic acid and were stirred at 500 rpm for 120 min until a homogeneous dispersion of the Al_2_O_3_ nanoparticles was formed. The Al_2_O_3_ dispersion solution was added to the PVDF solution for the nanocomposites. To obtain the final complex solution, vigorous mixing with a magnetic stirrer and gentle heating was used. The complex solution was then sprayed on the cleaned substrates using a spray coating process to create a flat surface. The coated substrates were then dried in a drying furnace at 80 °C for 10 min. Table 1 summarizes the trial conditions.

### 2.3. Characterization Techniques

The sessile droplet method was used to measure the water contact angle (WCA) and water contact angle hysteresis (WCAH) at room temperature using an Attension Biolin device (Model: Theta Optical Tensiometers, Helsinki, Finland). On the upper surface of the coated substrates, WCA and WCAH were measured using 5 µL of distilled water droplets. One sample was averaged from at least five independent determinations made at separate locations. The corrosion behavior of the coated steel and aluminum substrates was studied using VersaSTAT 3 Potentiostat/Galvanstate(AMETEK GmbH, Hadamar-Steinbach, Germany), by measuring the Tafel polarization curves. All of the corrosion tests were conducted at room temperature in 3.5 wt% NaCl solutions at a scan rate of 2 mVs^−1^ with a three-electrode cell consisting of a working electrode (bare or coated steel and aluminum substrates), a saturated calomel reference electrode (SCE), and a graphite counter electrode. An amount of 6.25 cm^2^ of the test sample was exposed to the electrolyte. The DeFelsko Digital Pull-off Adhesion Tester (Model: PosiTest AT-M, DeFelsko Corporation, Shandong, China) was used to measure the adhesion force between the coating layer and the substrates. A selective adhesive (epoxy) (ResinLab L.L.C., Germantown, MD, USA) was used to adhere each sample to 20 mm diameter dollies, after which the combination (dolly + sample) was placed inside the tester, and a force was applied to separate them. A scanning electron microscope (SEM) and energy dispersive spectroscope (EDS) (SEM Model: JSM-IT200, JEOL Ltd, Tokyo, Japan) with an accelerating voltage of 25 kV, were used to examine the morphology of the prepared membranes. Samples were vacuum dried and sputter-coated with Au before SEM examinations. Micrographs of the membrane surfaces were taken at various magnifications. To examine the chemical constituents of the samples, they were exposed to EDS to produce the most general representative spectra for each sample. The indentation hardness of the PVDF and PVDF/Al_2_O_3_ coatings was determined using the nanoindentation method (IIT) using a pyramidal-shaped Berkovich indenter (Nano indenter model: G200, KLA Corporation, CA, USA). A force of 0.15 mN was applied to accomplish the test.

## 3. Results and Discussion

### 3.1. Wettability Analysis

Contact angles are crucial indicators of solid surface energy since they are used to monitor the phase contact intensity between liquid and solid substrates. The contact angle hysteresis of the best conditions was also measured, which is defined as the angle at which a water droplet of a specific volume starts to slide down an inclined surface. Coating the fluorinated methyl groups onto flat solid surfaces results in a maximum WCA of just 120°, which is barely superhydrophobic [39]. Thus, to create superhydrophobicity, Al_2_O_3_ nanoparticles were applied to the solid surface. Figure 1, Figure 2 and Figure 3 show the average WCAs and WCAH of the PVDF–Al_2_O_3_ composites.

The WCA on bare steel was around 54 ± 3°, and 90 ± 2° after being coated with PVDF alone, while it increased to a maximum of 157 ± 2° after the addition of Al_2_O_3_ nanoparticles at an amount of 1.5 g, as shown in Figure 1. WCA on the bare Al substrates was 71 ± 3° and 91 ± 2° after coating them with PVDF alone, while it reached a maximum of 156 ± 3° with the addition of 1.5 g of Al_2_O_3_ nanoparticles, as shown in Figure 2. On the glass substrates, the WCA was around 51 ± 3° before coating and 93 ± 2° after coating it with PVDF alone, while it recorded a maximum of 148 ± 3° after the addition of Al_2_O_3_ nanoparticles, with an amount of 1.5 g, as shown in Figure 3. The greatest water contact angle hysteresis (WCAH) was achieved with 1.5 g of Al_2_O_3_ coatings on all substrates and ranged from 2 ± 1° to 8 ± 1° under such conditions. This means that adding Al_2_O_3_ to the PVDF improves its superhydrophobicity and that the best amount to use is 1.5 g of Al_2_O_3_ per 100 mL of the complex solution. With the addition of Al_2_O_3_, the observed rise in the water contact angle and the decrease in the contact angle hysteresis can be explained as follows: The number of nano-size asperities formed on the surface grew as the quantity of the Al_2_O_3_ addition increased, increasing the film roughness. These nano-sized asperities trap air and form air pockets between the water droplet and the gap between the asperities, resulting in a limited liquid–solid contact area for the water droplet. This finding has some relation to the well-known Cassie–Baxter theory [1]. The WCA of air was commonly thought to be 180°. According to Cassie–Baxter’s hypothesis, the Yong–Laplace pressure between the interfaces prevents the liquid droplet from making complete contact with the entire solid surface, so the asperities trap air inside, establishing a stable solid–air–liquid three-phase interface [40].

Equation (1) shows the relationship between the WCA on a flat surface (θ) and a rough surface (θ′) composed of a solid and air [41]:(1)$$\cos\left(\theta^{\prime }\right)=f_{1}\;\cos\left(\theta \right)-{\;f}_{2}$$
where f_1_ and f_2_ represent ratios of the solid surface and air in contact with liquid, respectively, in this equation. Given the WCAs of the flat PVDF film (∼90°) and the PVDF nanocomposite (∼157°), f_2_ was determined to be 0.92, indicating that the air trapped in the rough hierarchical micro-/nanostructures of Al_2_O_3_ was the primary cause of superhydrophobicity in the PVDF nanocomposites. As a result, after the three phases were stable, the higher proportion of trapped air would cause a higher WCA.

The water contact angle hysteresis (WCAH) as well as the static WCA are essential factors in defining surface hydrophobicity. The WCAH of the coating layer on the steel substrates, for example, was reduced from 38 ± 3° in the case of the PVDF-alone coating to only 7 ± 1° after 1.5 g of Al_2_O_3_ nanoparticles were added (Figure 1), allowing the water droplets to readily roll off of the surface. As a result, increasing the Al_2_O_3_ content in the PVDF polymer by up to 1.5 g/100 mL of the complex solution enhanced the hydrophobicity of the nanocomposite.

### 3.2. Corrosion Analysis

Tafel polarization curves were used to determine the corrosion resistance of the PVDF and PVDF/Al_2_O_3_ composite coatings. Figure 4 and Figure 5 show the Tafel polarization curves of the bare substrates, PVDF-coated substrates, and PVDF–1, 1.5, and 2 g Al_2_O_3_-coated substrates in 3.5 wt.% NaCl solution at a scan rate of 2 mVs^−1^ for the steel and Al substrates, respectively. Table 2 summarizes the Tafel analysis for the polarization curves.

The corrosion current density of the PVDF alone-coated steel and (PVDF–1.5 g Al_2_O_3_) coated steel was dramatically reduced from 37.75 in bare steel to 3.34 and 0.25 µAcm^−2^ in PVDF alone-coated steel and in PVDF–1.5 g Al_2_O_3_-coated steel, respectively, as seen in Table 2. Whereas, the corrosion current density was dramatically reduced from 8.26 in the case of bare Al to 1.3 and 0.15 µAcm^−2^ in the case of PVDF alone-coated Al and (PVDF–1.5 g Al_2_O_3_)-coated Al, respectively. Furthermore, the steel protection efficiency was improved from 91% with the PVDF only coating to 99% with the modified nanocomposite coating (PVDF–1.5 g Al_2_O_3_). The modified nanocomposite coating (PVDF–1.5 g Al_2_O_3_) raised the protective efficiency of Al from 84% in the case of the PVDF only coating to 98% in the case of the PVDF–1.5 g Al_2_O_3_ coating. Equation (2) was used to compute the protection efficiency for the coatings (η) [42]:(2)$$\eta =\frac{i_{1}-{\;i}_{2}}{i_{1}}\times 100\%$$
where i_1_ and i_2_ are the corrosion current densities of the bare substrates and coated ones, respectively.

Table 2 also shows the polarization resistance valu, R_p_, which was calculated using the Stern–Geary equation shown below:(3)$$R_{P}=\frac{\beta_{a}\beta_{c}}{2.303(\beta_{a}+\beta_{c})i_{\mathrm{corr}}}$$
where β_a_ and β_c_ are the anodic and cathodic Tafel slopes, respectively. i_corr_ is the corrosion current density, and R_p_ is the polarization resistance. Comparing the R_p_ of the (PVDF–1.5 g Al_2_O_3_)-coated steel to that for the PVDF alone-coated steel, it can be observed that the R_p_ for the case of the superhydrophobic PVDF–1.5 g Al_2_O_3_ composite coating is roughly one order of magnitude higher than that for the PVDF alone-coated one.

This was also the case for the Al substrates. Upon comparison of the R_p_ of the (PVDF–1.5 g Al_2_O_3_)-coated Al to that for the PVDF alone-coated Al, it can be shown that the R_p_ for the case of the superhydrophobic PVDF–1.5 g Al_2_O_3_ composite coating is two orders of magnitude higher than that for the PVDF only coated one. This indicates that both steel and Al, when coated with the PVDF–1.5 g Al_2_O_3_ composite, are significantly less susceptible to corrosion than in cases where they are covered with the PVDF alone.

Due to the presence of trapped air, the addition of Al_2_O_3_ nanoparticles to the PVDF polymer resulted in the creation of gaps at the polymer surface that hindered the entry of aggressive ions into the nanocomposite coating [43]. When only PVDF was utilized, however, the wide pore with low pore resistance did not prevent the aggressive ions from reaching the Al surface.

It can be concluded that the corrosion potential increases as the contact angle increased, and more hydrophobicity equaled higher corrosion resistance. From the obtained results, it is obvious that the various superhydrophobic PVDF–Al_2_O_3_ coatings outperform the hydrophobic PANI coatings in terms of anticorrosion performance.

### 3.3. Adhesion Analysis

The adhesion force of the produced coatings is shown in Figure 6. The adhesion force values for each type of substrates follow a similar approach in coated materials, where the coatings of PVDF alone have the largest adhesion with the substrates. This adherence is reduced when Al_2_O_3_ nanoparticles are added. Coatings with low levels of nanoparticles (1 g Al_2_O_3_) have a higher adhesion force, whereas coatings with a high Al_2_O_3_ content have a lower adhesion force.

The cross-linking effect of the PVDF macromolecules via the production of C–C or C=C functional groups during the drying process is primarily responsible for the high adhesion force of the interface of the produced coatings [19]. The addition of Al_2_O_3_ allows some nanoparticles to exist at the interface between the coating and the substrate, which reduces the contact area of the substrate’s surfaces with the highly adhesive PVDF. The PVDF-exposed area of the substrate shrinks as the Al_2_O_3_ content in PVDF + Al_2_O_3_ coating increases. As a result, increasing the amounts of Al_2_O_3_ nanoparticles leads to a decrease in the adhesion between the coating and the substrate. According to the above-mentioned WCA and adhesion results, coatings of PVDF + 1.5 g Al_2_O_3_ are acceptable in terms of adhesive superhydrophobic characteristics since this condition combines superhydrophobicity with moderate adhesion.

### 3.4. Surface Morphology Analysis

In defining the wetting properties of the coated surfaces, the surface micro/nanostructure received a lot of attention. Superhydrophobicity is mostly caused by micro- and nanoscale hierarchical surface patterns where low surface energy is responsible for superhydrophobicity.

The morphology of the produced coatings surface was studied using SEM at different magnifications. The average thickness of the coated material was 1.49 ± 0.04µ. At high magnification, the hydrophobic surface microstructure depicted in Figure 7a–c displays sponge-like structures, indicating that the coating film is primarily made of PVDF polymer [27,44]. The hydrophobic PVDF sample has a uniform surface morphology, as seen in Figure 7a. Beyond these formations, there are several huge microvoids. EDS was utilized to confirm the composition of the PVDF polymer, as shown in Figure 7c. The PVDF material comprises two principal peaks of fluorine (F) and carbon (C), according to EDS measurements.

At high magnification, the hydrophobic surface microstructure depicted in Figure 7a–c displays sponge-like structures, indicating that the coating film is primarily made of PVDF polymer. Beyond these formations, there are several huge microvoids. EDS was used to confirm the composition of PVDF polymer, as shown in Figure 7c. The PVDF material comprises two principal peaks of fluorine (F) and carbon (C), according to EDS measurements.

The addition to the Al_2_O_3_ nanoparticles resulting in a noticeable change in morphology, as shown by the bright spots in Figure 8a–c, as shown in Figure 8a, the composite coatings contained Al_2_O_3_ particle aggregates. The micro–nano scale-structured roughness surfaces were produced using Al_2_O_3_ particles with a range of morphologies, but these were mostly agglomerates because the size was less than 100 nm, as shown in Figure 8c. The superhydrophobicity of the films with a static WCA of 157 ± 2° and a WCAH of 8 ± 1° is aided by this micro–nano scale structure. The principal chemical components of the samples containing PVDF + 1.5 g Al_2_O_3_ are shown in the EDS spectrums in Figure 8d. Carbon and fluorine were found in the PVDF structure, whereas aluminum (Al) and oxygen (O) were found in the integrated Al_2_O_3_. Table 3 shows the elemental analysis of the coated surfaces using EDS.

### 3.5. Nanohardness Analysis

Nanoindentation tests were performed on the PVDF only coatings together with those containing 1.5 g of Al_2_O_3_ nanoparticles. The results revealed that the addition of Al_2_O_3_ nanoparticles enhanced both the hardness and elastic modulus of the coatings. Figure 9 shows that the depth of penetration of the indenter for the PVDF only coatings largely decreased after introducing the Al_2_O_3_ nanoparticles.

As shown in Table 4, the addition of Al_2_O_3_ nanoparticles increased the value of the elastic modulus from 789 MPa in the case of the PVDF only coating to 1.073 GPa. Additionally, the hardness value of PVDF alone-coatings jumped from 105 MPa to 190 MPa. The relatively high hardness of Al_2_O_3_ nanoparticles played a significant role in enhancing the mechanical properties of the PVDF coatings.

## 4. Conclusions

The conclusions based on the foregoing discussion can be described as follows:

Spray coating methods were successfully used to make PVDF/Al_2_O_3_ composites. This technology could be used in a large-scale procedure and could be a cost-effective option for industrial and civil applications.

The low surface energy of the PVDF, as well as the mixture of hierarchical micro and nanostructures of Al_2_O_3_ embedded in the polymer, are responsible for the superhydrophobicity of PVDF/Al_2_O_3_ composite coatings.

The increased concentration of Al_2_O_3_ nanoparticles in the PVDF matrix up to 1.5 g/100 mL of complex solution on steel substrates resulted in a substantial increase in WCA from 90 ± 2° to 157 ± 2°; however, the WCAH decreased to 8 ± 1°. Similar results were obtained for both the Al and glass substrates.

Al_2_O_3_ nanoparticles improve the hydrophobicity of PVDF coatings by reducing the pore size inside the composite coating and by increasing air trapped within the interstices of the surface.

The corrosion resistance of the superhydrophobic PVDF/Al_2_O_3_ composite coatings was significantly improved since the corrosion rates were 154 and 59 times lower than the bare steel and Al substrates, respectively, and 70 and 44 times lower than the corresponding prepared PVDF coating without Al_2_O_3_ nanoparticles on the steel and Al substrates, respectively).

Although the adhesion force of PVDF alone is higher than that of the PVDF + 1.5 g Al_2_O_3_ composite coatings, the adhesion of the composite is fairly accepted.

Both hardness and elastic modulus values of the composite coatings (PVDF + 1.5 gm Al_2_O_3_) were increased by 81% and 36%, respectively, compared to PVDF alone.

The overall results suggest that the prepared PVDF–Al_2_O_3_ composite coating can be used to develop excellent superhydrophobic surfaces that might be potentially useful for various applications.

## Figures and Tables

**Figure 1 materials-14-06358-f001:**
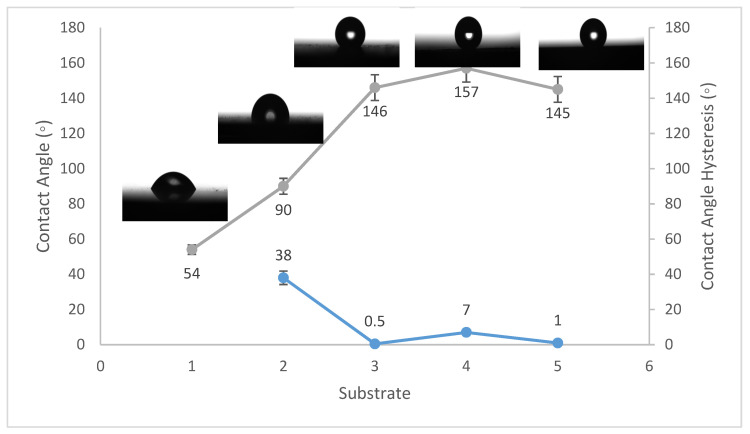
Wettability (WCA and WCAH) of S1, S2, S3, S4, and S5 conditions (steel substrates).

**Figure 2 materials-14-06358-f002:**
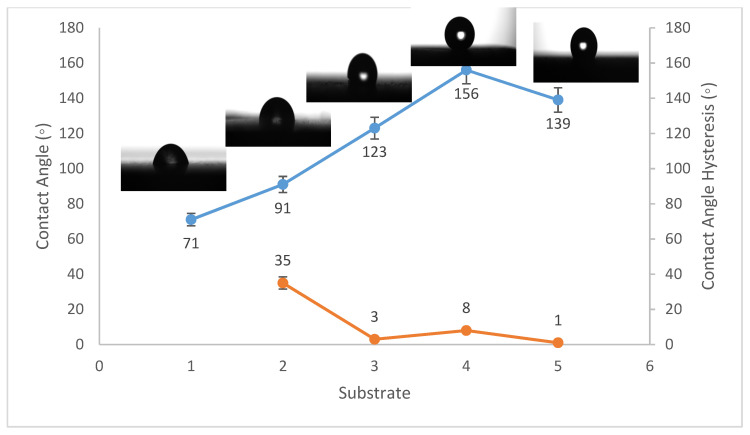
Wettability (WCA and WCAH) of A1, A2, A3, A4, and A5 conditions (Al substrates).

**Figure 3 materials-14-06358-f003:**
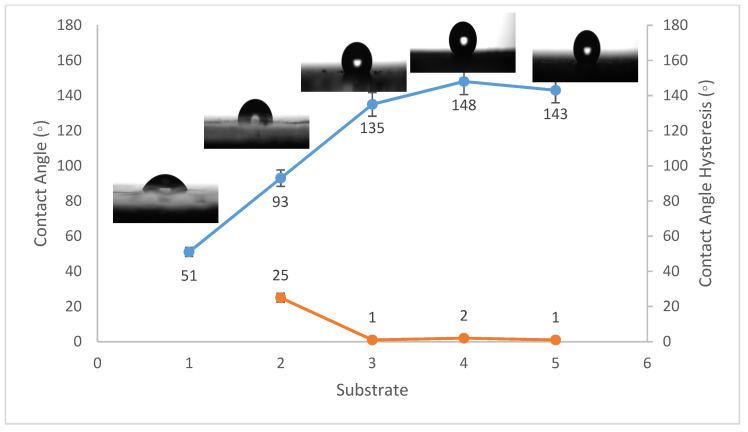
Wettability (WCA and WCAH) of G1, G2, G3, G4, and G5 conditions (glass substrates).

**Figure 4 materials-14-06358-f004:**
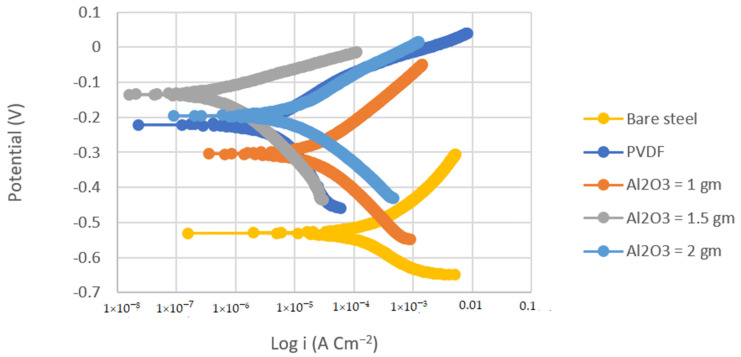
Tafel plots of bare steel, PVDF-coated steel, and coated steel with coatings of different Al_2_O_3_ contents (1, 1.5, and 2 g) in 3.5% NaCl solution (Scan rate = 2 mVs^−1^).

**Figure 5 materials-14-06358-f005:**
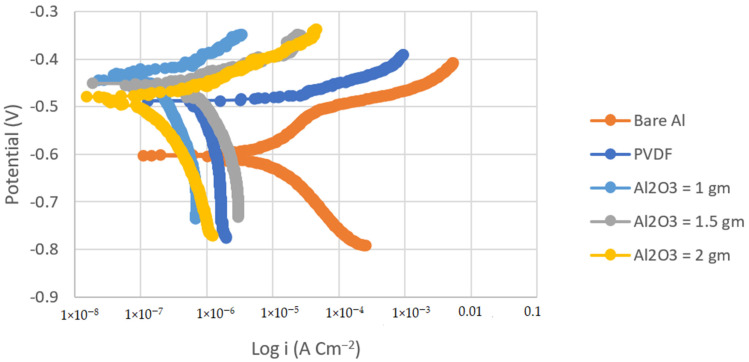
Tafel plots of bare Al, PVDF-coated Al, and coated Al with coatings of different Al_2_O_3_ contents (1, 1.5, and 2 g) in 3.5% NaCl solution (Scan rate = 2 mVs^−1^).

**Figure 6 materials-14-06358-f006:**
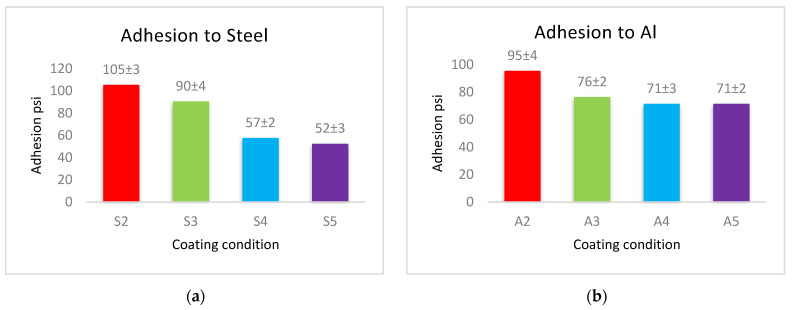
The adhesion force between (**a**) steel, (**b**) Al, and (**c**) glass substrates, and PVDF alone and PVDF–Al_2_O_3_ coatings.

**Figure 7 materials-14-06358-f007:**
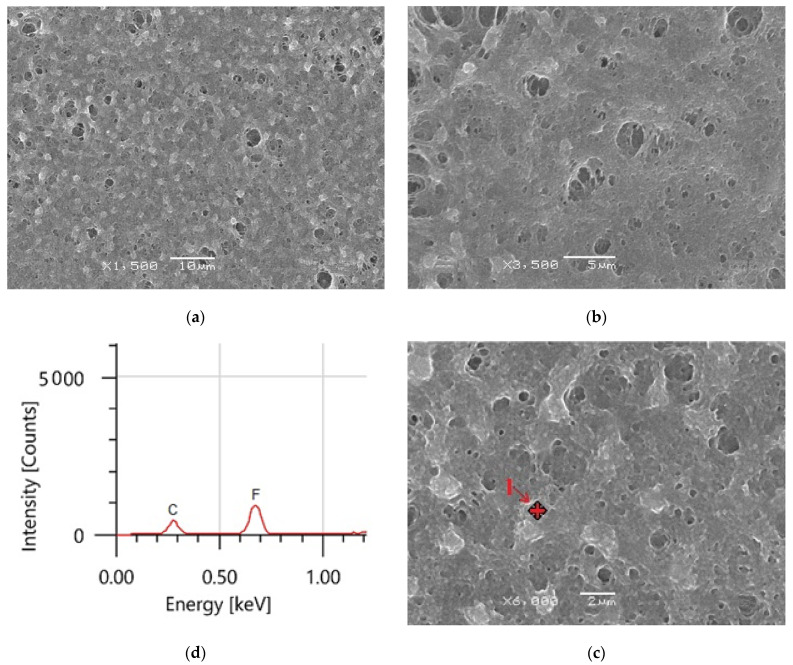
(**a**–**c**) PVDF alone membrane images with different magnifications and (**d**) EDS analysis of point 1 in (**c**).

**Figure 8 materials-14-06358-f008:**
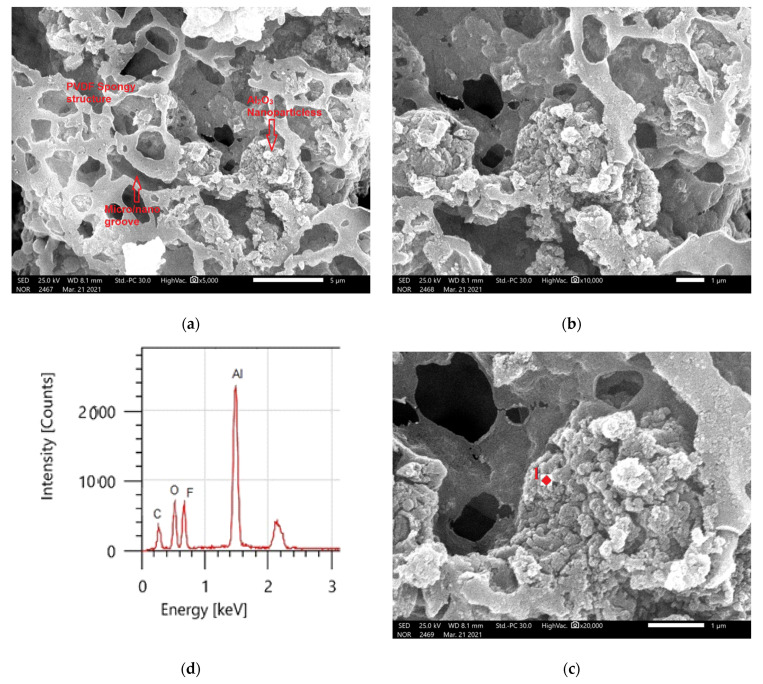
(**a**–**c**) PVDF composites containing 1.5 g Al_2_O_3_ at different magnifications, as shown by SEM and (**d**) EDS analysis of point 1 in (**c**).

**Figure 9 materials-14-06358-f009:**
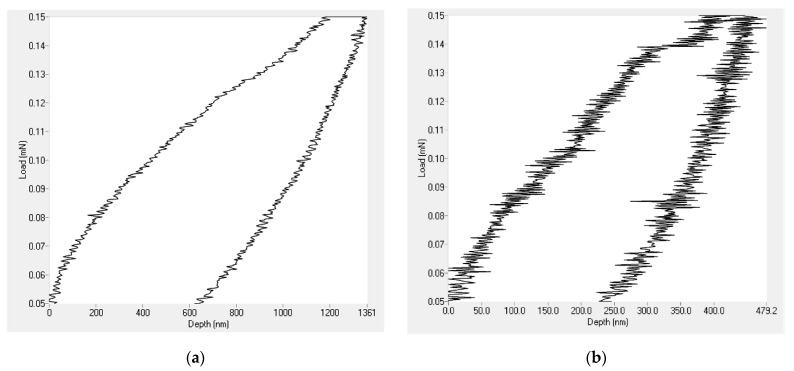
Nanoindentation loading and unloading curve for (**a**) PVDF only coating and (**b**) PVDF composites containing 1.5 g Al_2_O_3_.

**Table 1 materials-14-06358-t001:** Experimental conditions.

Substrate	Condition	Amounts per 100 mL of Complex Solution	Substrate No.
Steel	Bare	-	S1
	PVDF	5.0 g	S2
	Al_2_O_3_	1.0 g	S3
		1.5 g	S4
		2.0 g	S5
Al	Bare	-	A1
	PVDF	5.0 g	A2
	Al_2_O_3_	1.0 g	A3
		1.5 g	A4
		2.0 g	A5
Glass	Bare	-	G1
	PVDF	5.0 g	G2
	Al_2_O_3_	1.0 g	G3
		1.5 g	G4
		2.0 g	G5

**Table 2 materials-14-06358-t002:** Tafel analysis for bare substrates, PVDF alone, and (PVDF–1.5 g Al_2_O_3_) composite coatings after being tested into 3.5% NaCl aqueous solution. The scan rate was 2 mVs^−1^.

Steel Substrates	β_a_ (mV)	β_c_ (mV)	R_p_ (KΩ cm^2^)	CR Rate (mpy)	i_corr_ (µA/cm^2^)	η (%)
Bare Steel	117	139	0.7	17	37.75	-
PVDF	73	198	6.9	0.24	3.34	91
PVDF + 1.5 g Al_2_O_3_	43	157	58.6	0.11	0.25	99
**Al Substrates**						
Bare Al	54	122	1.9	3.54	8.26	-
PVDF	93	865	28	0.08	1.3	84
PVDF + 1.5 g Al_2_O_3_	39	467	292.8	0.06	0.15	98

**Table 3 materials-14-06358-t003:** EDS analysis of surface elements for PVDF alone and PVDF–Al_2_O_3_ composite coatings on Al substrate.

Element	PVDF Alone		PVDF + 1.5 g Al_2_O_3_	
	Mass %	Atom %	Mass %	Atom %
C	47.65	59.01	28.47	39.14
F	52.35	40.99	27.16	23.60
O			24.05	24.82
Al			20.32	12.44
Total	100.00	100.00	100.00	100.00

**Table 4 materials-14-06358-t004:** Hardness and elastic modulus values for coatings of both PVDF alone and PVDF + 1.5 g Al_2_O_3_.

Sample	PVDF Alone	PVDF + 1.5 g Al_2_O_3_
Hardness (MPa)	105	190
Elastic Modulus (MPa)	789	1073
Maximum Depth (nm)	1360	479

## Data Availability

Data available on request from the corresponding author.

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
