# Peer review of "Performance of Sprayed PVDF-Al2O3 Composite Coating for Industrial and Civil Applications"

_materials, 2021, doi:10.3390/ma14216358_

Round 1

Reviewer 1 Report

In this work, PVDF/Al2O3 composite coatings were prepared using spray technique and the wettability and corrosion behaviors of the coated materials were investigated.

The reviewer has the following comments and criticisms on this manuscript.

1. Overall, innovation of the work is not obvious.
2. In Abstract section, main findings rather than tests should be highlighted.
3. The authors should carefully check the accuracy of the description and analyses in the text. For example:
1) On Page 4: “contact angles are the distinctive constants of …”. Contact angle is not a constant.
2) On Page 4: “The sliding angle of the best conditions was …”. However, no sliding angle data were provided in Results section; Moreover, “of the best conditions”?
3) On Page 5: “The greatest water contact angle hysteresis (WCAH) was achieved with 1.5 g of Al2O3 coatings on all substrates, ranged from 2±1◦ to 8±1◦ under such conditions. This means that adding Al2O3 to the PVDF improves its superhydrophobicity, …”. Why was the largest water contact angle hysteresis mentioned?

Superhydrophobic surfaces should have high contact angles and low contact angle hysteresis.
4) On Page 6: “… to only 8±1◦ after 1.5 g of AlO3 nanoparticles were added (Figure1), …”. Here, “8±1◦” should be “7±1◦”.
5) On Page 9: “The hydrophobic sample has a homogeneous surface structure and minor roughness, which should be observed”. “homogeneous”? “minor”? “should be observed”?
6) On Page 10: “At high magnification, the hydrophobic surface microstructure … PVDF material comprises principally two peaks of fluorine (F) and carbon (C), according to EDS measurements”. Repetition.
7) On Page 11: “The superhydrophobicity of the films with a static WCA of 157±2◦ and a SA of 8±1◦ is aided by this micro-nano scale structure”. However, no SA (sliding) was provided.

4. The language should be improved in some places.

Author Response

The authors thank the reviewer for reviewing the manuscript and providing constructive comments to improve the quality of the manuscript. The authors have tried their best to respond to the comments of the reviewers and hope the reviewers are satisfied with the responses. All modifications were highlighted in yellow color.

Reviewer 2 Report

The subject matter is interesting, however, there are still some things that could be improved, and a few questions that have to be answered before publication. Therefore, I suggest a mandatory revision of the following points to increase the quality of the paper:
1. Why did the authors produce VDF/Al2O3 composite coatings on the glass? There is no information on this in the introduction.
2.  There are no parameters of SEM, for example, accelerating voltage that influence on the EDS microanalysis resolution.
3. The authors should show pictures of the nanoparticles.
4. Did the use of Al2O3 particles in the form of nanoparticles make sense according to the authors, if Al2O3 conglomerates in the coating were obtained anyway?
5. What thicknesses were the coatings?
6. The authors should show SEM images of coatings in cross-section.
7. They should show the distribution of nanoparticles in the coating.
8. The authors should show SEM pictures of the places where the hardness tests were performed. Tests at such low loads are erroneous and concern very small areas of the microstructure. The authors should perform hardness tests with higher loads. Why used such a low force?
9. Why did the authors perform the corrosion resistance in a 3.5% NaCl solution? It requires comments.
10. The discussion of the results is quite weak, especially regarding the corrosion resistance.

Recommendation:
This manuscript in the presented form is not acceptable for publication in the Materials. The major revision is necessary.

Author Response

(The authors gave the same response as above.)

Round 2

Reviewer 2 Report

The authors have not completely taken into account the reviewer's comments and they have not made corrections in this article. Due to the high quality of the Materials, additional research is necessary. In my opinion, it's unreasonable that the authors have written: "the SEM lab is very busy and they asked us to wait at least one month....  The journal gave us only 10 days for reply". In my opinion, authors should ask the editor to increase the submission time of the revised manuscript in order to perform the necessary research.

The authors have answered the question about the thickness of the coatings. By what method was the thickness of the coatings measured? They should provide proof in the form of pictures of the coatings. 

Recommendation:
This manuscript in the presented form is not acceptable for publication in the Materials. The major revision is necessary.

Author Response

The authors thank the reviewers for reviewing the manuscript and providing constructive comments to improve the quality of the manuscript. The authors have tried their best to respond to the comments of the reviewers and hope the reviewers are satisfied with the responses.
